# Dietary Exposure to Polychlorinated Biphenyls and Dioxins and Its Relationship to Telomere Length in Subjects Older Than 55 Years from the SUN Project

**DOI:** 10.3390/nu14020353

**Published:** 2022-01-14

**Authors:** Lucia Alonso-Pedrero, Carolina Donat-Vargas, Maira Bes-Rastrollo, Ana Ojeda-Rodríguez, Guillermo Zalba, Cristina Razquin, Miguel A. Martínez-González, Amelia Marti

**Affiliations:** 1Department of Nutrition, Food Science and Physiology, University of Navarra, 31008 Pamplona, Spain; lalonso.1@alumni.unav.es (L.A.-P.); aojeda.5@alumni.unav.es (A.O.-R.); 2Navarra Institute for Health Research (IdiSNA), 31008 Pamplona, Spain; mbes@unav.es (M.B.-R.); gzalba@unav.es (G.Z.); mamartinez@unav.es (M.A.M.-G.); 3Departament of Preventive Medicine and Public Health, School of Medicine, Universidad Autónoma de Madrid, CEI UAM+ CSIC, 28049 Madrid, Spain; carolina.donat.vargas@ki.se; 4Unit of Nutritional and Cardiovascular Epidemiology, Environmental Medicine Institute (IMM), Karolinska Institutet, 17177 Stockholm, Sweden; 5Physiopathology of Obesity and Nutrition Networking Biomedical Research Centre (CIBERobn), Spanish National Institute of Health Carlos III, 28029 Madrid, Spain; crazquin@unav.es; 6Department of Preventive Medicine and Public Health, University of Navarra, 31008 Pamplona, Spain; 7Department of Biochemistry and Genetics, University of Navarra, 31008 Pamplona, Spain; 8Department of Nutrition, Harvard TH Chan School of Public Health, Boston, MA 02115, USA

**Keywords:** polychlorinated biphenyls (PCBs), dioxins, telomere length (TL), cross-sectional study, Seguimiento Universidad de Navarra (SUN) study, nutritional epidemiology

## Abstract

Exposure to persistent organic pollutants (POPs) may influence telomere length (TL), which is considered as a marker of biological age associated with the risk of chronic disease. We hypothesized that dietary exposure to polychlorinated biphenyls (PCBs) and dioxins could affect TL. Our aim was to evaluate the association of dietary exposure to PCBs and dioxins with TL. In this cross-sectional study of 886 subjects older than 55 y (mean age: 67.7; standard deviation (SD): 6.1; 27% women) from the “Seguimiento Universidad de Navarra” (SUN) project. TL was determined by real-time quantitative polymerase chain reaction and dietary PCBs and dioxins exposure was collected using a validated 136-item Food Frequency Questionnaire. Multivariable linear regression models were used to control for potential confounding factors. Shorter TL was associated with dietary total PCBs (SD of T/S ratio/(ng/day) = −0.30 × 10^−7^; 95% CI, −0.55 × 10^−7^ to −0.06 × 10^−7^), dioxin-like PCBs (DL-PCBs) (SD of T/S ratio/(pg WHO TEQ (Toxic Equivalents)/day) = −6.17 × 10^−7^; 95% CI, −11.30 × 10^−7^ to −1.03 × 10^−7^), and total TEQ exposure (SD of T/S ratio/(pg WHO TEQ/day) = −5.02 × 10^−7^; 95% CI, −9.44 × 10^−7^ to −0.61 × 10^−7^), but not with dioxins (SD of T/S ratio/(pg WHO TEQ/day) = −13.90 × 10^−7^; 95% CI, −37.70 × 10^−7^ to 9.79 × 10^−7^). In this sample of middle-aged and older Spanish adults, dietary exposure to total PCBs and DL-PCBs alone and together with dioxins was associated with shorter TL. Further longitudinal studies, preferably with POPs measured in biological samples, are needed to confirm this finding.

## 1. Introduction

Telomeres are 5–10 kb tandemly TTAGGG repeated sequences located at chromosomes ends, whose main function is chromosome stability and integrity [1]. Telomeres are a heritable factor considered as a biological clock since they decrease their length over time, in other words, with every cell division [1]. However, several behavioral and environmental exposure factors may increase their speed of shortening, such as smoking, obesity, unhealtjy diet, stress [1], and pollution [2].

Among possible contributors to changes in telomere length (TL), persistent organic pollutants (POPs) found in the environment, water, and food have recently gained importance. polychlorinated biphenyls (PCBs) and dioxins are chlorinated chemicals that belong to the group of environmental pollutants classified as POPs, a group of toxic chemical compounds very resistant to degradation and with high lipophilic and bio-accumulative potential (in adipose tissue of living organisms) [3]. Because of these characteristics, even though they were banned 40 years ago in many countries, they are still present in the environment and food supply on a global scale [4], food being the main source of human intake [5]. Interestingly, because they are eliminated very slowly, the body burden accumulated during the whole lifetime is high [6]. PCBs were used as industrial fluids, including dielectric coolants in capacitors and transformers [4]. An important number of observational studies have shown detrimental associations between several POP compounds and high risk of chronic diseases, including type 2 diabetes [7], obesity [8], hypertension [9,10], and cardiovascular diseases [6,11]. Likewise, dioxins, which include polychlorinated dibenzodioxins (PCDDs) and polychlorinated dibenzofurans (PCDFs), are unintentionally compounds produced as industrial bioproducts in several industrial settings. The most relevant outcome of dioxins at high doses both in humans [12] and animals [13] is cancer.

Recent studies showed that PCBs are likely to regulate telomere length and telomerase activity. Whereas six population-based studies showed that serum levels of POPs were associated with longer telomeres [14,15,16,17,18,19], several studies showed shorter TL associated with POPs when they were analyzed in immortal human skin keratinocyte (HaCaT) cells [20,21,22] and undifferentiated human myeloblastic/promyelocytic leukemia cell line (HL-60) [23]. Moreover, an inverse association between COPs and LT in lymphocytes [4,24] and buccal cells [25] was found. The mechanisms underlying these associations are as yet unknown; however, it seems that POPs produce specific adverse effects on TL, hTERT activity, and expression of telomere-associated shelterin genes (TRF1/2, POT1).

Since PCBs and dioxins have been reported to be associated with chronic diseases, our interest was to evaluate their relationship with TL. Therefore, we conducted a cross-sectional analysis with the aim of examining whether dietary exposure to PCBs and dioxins may be associated with TL in a subset of subjects over 55 years from the SUN study.

## 2. Materials and Methods

### 2.1. Study Sample

The present study included a sample of participants from the Seguimiento Universidad de Navarra (SUN) study, a prospective and dynamic cohort of Spanish university graduates [26] (ClinicalTrials.gov, Identifier: NCT02669602). The recruitment of this cohort started in 1999, and the enrollment is permanently open. Briefly, baseline and follow-up questionnaires are mailed every 2 years to follow-up the participants. More details on the recruitment and follow-up methods are described elsewhere [26].

A genetic study was performed within the SUN cohort in May 2008, where 1921 adults over 55 years were invited to participate. From those, 1085 accepted to participate and 986 returned saliva samples, even though only 953 samples could be correctly analyzed. Furthermore, we excluded 67 participants reporting a total energy intake outside Willett’s predefined values (<800 or >4000 kcal/day for men and <500 or >3500 kcal/day for women), leaving a study population of 886 individuals (Figure 1). The study protocol was written in accordance with the principles of the Declaration of Helsinki and was approved by the University of Navarra institutional review board. Specific written informed consent was requested to participate in this study.

### 2.2. Assessment of Dietary Exposures to PCBs and Dioxins

Daily dietary exposure to total PCBs (ng/day), Dioxin-like PCBs (DL-PCBs) (pg WHO (World Health Organization), TEQ (Toxic Equivalents)/day), total dioxins, and total exposure (DL-PCBs and dioxins) (pg WHO TEQ/day) was estimated by the average concentration in various foods, obtained from a previous study performed in Spain [27], with the respective frequency of consumption (in grams) obtained from the baseline 136-item semiquantitative Food frequency questionnaire (FFQ) previously validated [28].

The estimated total PCBs, DL-PCBs, total dioxins, and total exposure (DL-PCBs and dioxins) was adjusted for total energy intake (mean of 2247.493 kcal per day for this sample) through the residuals method [29], since the total caloric intake can be considered as a potential confounding factor for the studied association. This way we could ensure that the association between TL and dietary chlorinated POPs exposure was completely independent of total energy intake.

The percentage of the main food groups contributing to the total amount of PCBs, dioxins, and total exposure (DL-PCBs + dioxins) consumed in participants from the SUN study is shown in Appendix A. We calculated it by dividing the amount of each food group in grams (multiplied by the average concentration obtained from a previous study [27]) by the total grams of them and multiplying by 100. The main food source of PCBs and total dietary exposure (DL-PCBs + dioxins) in this study was fatty fish, as previously reported [8,9]. Furthermore, the highest contribution of dioxins, in addition to fatty fish, was lettuce, which contributed almost 10% of their total dietary exposure.

### 2.3. Outcome Assessment: TL

Purified DNA was extracted from saliva samples (collected with DNA Collection—Oragene OG-250 Saliva Kit) with the use of a DNA blood extraction kit (Pure Link Genomic DNA, Invitrogen) and frozen at −80 °C. TL was measured by a monochrome multiplex real-time quantitative PCR method (MMqPCR based on the Cawthon´s method [30]), as previously described in a CFX384 Touch-Real-time PCR system (BioRad) [31,32,33]. Briefly, this method performs in a single reaction the quantification of the relative copy numbers of telomeres and a single copy gene (albumin), and TL is expressed as a ratio of these two parameters (T/S ratio) [34]. The acceptable ranges of PCR efficiency for the single copy gene and telomere primers were 90 and 110%, respectively.

For quality control, each sample was run in triplicate. In addition, a seven-point standard curve made from reference DNA samples was included for each plate using a 2-fold dilution series of DNA, ranging from 150 to 234 ng/mL. TL was expressed as a T/S ratio using the calibration curve (linearity agreement R^2^ > 0.99) to relative quantification. The intra-class correlation coefficient for TL was 0.793 (95% CI: 0.707, 0.857).

### 2.4. Assessment of Other Variables

Baseline questionnaire of the SUN project also included information about other covariates previously validated in the cohort [26], including sociodemographic variables (age, sex, and educational level), anthropometric variables (weight and height), participants’ medical history (personal and family history of CVD, obesity, hypertension, diabetes, cancer, and dyslipidemia), and health-related habits [28] (smoking status, physical activity, computer hours, TV hours, sleeping hours, sleeping/siesta, snacking between hours, alcohol consumption, cholesterol intake, fiber intake, total fats intake, energy intake, ultra-processed food consumption, following a special diet, and Mediterranean diet).

### 2.5. Statistical Analyses

The baseline characteristics were determined according to the median of dietary PCBs and dioxins (PCBs: 1125.13 ng/day; Dioxins: 32.80 pg WHO (World Health Organization) TEQ/day; POPs: 97.80 pg WHO TEQ/day). Differences between groups (< or ≥ the median) were assessed using the Mann-Whitney U test for continuous variables and Pearson’s chi-squared test for categorical variables.

Spearman correlations were used as a quality control in our study to assess the association between TL and age at saliva collection. We also used Spearman correlation coefficients to further evaluate the associations between PCBs and dioxins.

Multivariate linear regression models were conducted to assess the relationship of dietary exposure to PCBs or dioxins with TL. We used the standard deviation (SD) of the T/S ratio because showing that there is a change of such magnitude for each SD of the independent variable gives a better idea of the effect. We ran four multivariable-adjusted models with progressive adjustment for covariates that may possibly confound the association. Model 1 was adjusted for age and sex. Model 2 was further adjusted for BMI (kg/m^2^), energy intake (kcal/day) and personal history of CVD, obesity, HTA, diabetes, cancer, and dyslipidemia (categorical). Model 3 was further adjusted for educational level (year at university, continuous), smoking status (current, never, former), physical activity (MET-h/week), hours of computer use, hours of TV watching, sleeping hours, sleeping/siesta (categorical), and snacking between hours (categorical). Model 4 was further adjusted for alcohol consumption (g/day), cholesterol intake (mg/day), fiber intake (g/day, continuous), total fats intake (percentage of total energy intake of lipids), ultra-processed food consumption (servings/day), following a special diet (categorical), and adherence to a Mediterranean diet (Trichopoulou’s 0 to 9 Mediterranean diet score).

We represented relative telomere length with vigintiles of DL-PCBs in a box plot. The coefficient of correlation was calculated using the inverse probability method adjusted for variables in Model 4.

As sensitivity analyses, we reran the analyses, additionally using the 5th and 95th percentiles as limits for allowable total energy intake, further adjusting for omega-3 fatty acids intake. Years elapsed between saliva collection and the inclusion in the study and excluded participants with family and personal history of obesity, cardiovascular disease, hypertension, diabetes and cancer, and participants taking drugs to treat Diabetes and immunosuppressants for Alzheimer’s.

All *p*-values were 2-tailed, and a *p*-value < 0.05 was considered statically significant. Analyses were performed with STATA version 12 (STATA Corp., College Station, TX, USA).

## 3. Results

The mean age of participants was 67.7 ± 6.1 years, and the mean body mass index was 25.9 ± 3.2 kg/m^2^. Table 1 shows the characteristics of participants according to the median of dietary PCBs (ng/day), dioxins (pg WHO TEQ/day), and total (pg WHO TEQ/day) exposure (DL-PCBs + dioxins). Participants above the median value of dietary PCBs, dioxins, and total exposure (DL-PCBs and dioxins) had higher adherence to the Mediterranean diet and greater intake of total cholesterol, fiber, carbohydrates, and proteins, but they exhibited lower values of ultra-processed food consumption.

Moreover, subjects above the median value of dietary dioxins exposure had higher total fat intake, lower alcohol consumption, and were more likely to be never smokers.

Spearman correlations between dietary exposures to the chlorinated POPs analyzed in this study as a quality control are shown in Figure 2. We observed that all the chlorinated POPs from this study were positively correlated with each other. 10,000.

Regarding TL, as expected, a significant correlation was observed between TL and age in our population (correlation coefficient = −0.197, *p* ≤ 0.001) (Appendix A). We assessed the association between TL and total PCBs, DL-PCBs, total dioxins, and total dietary exposure (DL-PCBs + dioxins) in multivariate linear regression models (Table 2). Interestingly, we found significant inverse associations between TL and total PCBs (standard deviation (SD) of T/S ratio/(ng/day): −0.30 × 10^−7^; 95% CI, −0.55 × 10^−7^ to −0.06 × 10^−7^; *p* value = 0.015), DL-PCBs (SD of T/S ratio/(pg WHO TEQ/day): −6.17 × 10^−7^; 95% CI, −11.30 × 10^−7^ to −1.03 × 10^−7^; *p* value = 0.019) and total dietary exposure (DL-PCBs and dioxins) (SD of T/S ratio/(pg WHO TEQ/day): −5.02 × 10^−7^; 95% CI, −9.44 × 10^−7^ to −0.61 × 10^−7^; *p* value = 0.026). Moreover, we represented a box plot of relative TL with vigintiles of DL-PCBs (correlation coefficient inverse probability weighting-adjusted vigintiles of DL-PCBs = −0.1896, *p*-value: <0.001) (Appendix A).

We conducted multiple sensitivity analyses to account for potential uncertainties (Table 3). The association between dietary exposure to total PCBs, DL-PCBs, total dioxins, and DL-PCBs + dioxins did not vary after changing the energy limits to the 5th and 95th percentile, after additional adjusting for omega-3 fatty acids intake, or after excluding participants with family history of obesity, hypertension, diabetes, or participants with a personal history of obesity, hypertension, diabetes, CVD, cancer, and taking drugs to treat Diabetes and immunosuppressants for Alzheimer’s.

## 4. Discussion

The results of this study of participants over 55 years from the SUN cohort showed a negative association between exposure to total PCBs, DL-PCBs, or total dietary exposure to these compounds (DL-PCBs and dioxins) and TL. However, no association between dietary exposure to dioxins and TL was observed. To the best of our knowledge, this is the first epidemiologic study in humans assessing chlorinated POPs through a validated FFQ in association with TL.

In some studies, long-term exposure to low doses of non-orthochlorinated PCBs, DL-PCBs, and NDL-PCBs measured in serum levels were associated with longer telomeres [14,15,16,19,35,36]. In contrast to these observations, in vitro studies with individual PCB congeners in air showed reduced hTERT enzyme activity when exposure to POPs was tested in HaCaT cell cultures [20,21,22] and HL-60 leukemic cells [23]. In HaCaT cell cultures, telomere shortening could be explained by telomerase inhibition [21]. In addition, Ziegler et al. found a very significant reduction in age-adjusted TL in lymphocytes from individuals exposed to high levels of PCBs [4]. Accelerated telomere shortening in lymphocytes could thus be contributed to the immunosuppressive activity of PCBs and may explain some of the long-term effects of PCB exposure on the adaptive immune system [24]. Likewise, in several studies evaluating exposure to pesticides [25,37], a decrease in TL was observed in DNA isolated from buccal cells.

These results are difficult to compare, firstly due to the diversity of studies regarding the evaluation of the contaminant (POPs) in a human biological fluid, such as serum, or using a validated questionnaire for the assessment of dietary intake and also in vitro studies. In addition, the dose of POPs, the duration of exposure, and the cells in which TL is measured, as well as other factors, for example related to lifestyle factors, may influence telomere homeostasis [38].

Dietary chlorinated POPs exposure was associated with several diseases increasing the risk of CVD. It was found within the SUN cohort that PCBs levels are associated with higher risk of developing hypertension [9] and higher incidence of obesity [8]. Moreover, PCBs were associated in other studies with coronary atherosclerosis [6], heart failure [11], type 2 diabetes [7], and cardiovascular risk mortality [3], among others.

Diet is the main route of PCBs, dioxins, and other POPs for human exposure (90%) [39]. We observed that fish was the main food group contributing to PCBs and dioxin exposure. Recent epidemiological studies found that omega-3 fatty acids consumption (which is the main source of the fish) is able to reduce cardiovascular mortality [3]. However, PCBs exposure (for which the main food contributor is fish) is linked to cardiovascular mortality [11]. This is a controversial scenario, and the justification could be that the benefits of fish itself seem to overcome the negative effects of PCBs. Moreover, the beneficial effect is observed for moderate consumption of fish, included in a Mediterranean diet-based pattern, which, in addition, is accompanied by other food items that can also counteract the effects of POPs on TL, such as fruits and vegetables. Notably, we did not observe changes in our results when we adjusted for the consumption of omega-3 fatty acids.

We also excluded participants with family and personal history of several diseases, as they also affect the homeostasis of TL in humans. In this regard, we excluded participants taking drugs to treat diabetes (metformin) and immunosuppressants (rapamycin). As it is known, Metformin does attenuate the hallmarks of aging, including telomere erosion [40]; meanwhile, Rapamycin, which provides neuroprotection for neurodegenerative diseases (Alzheimer’s, Parkinson’s, Huntington’s, and spinocerebellar ataxia type 3), through the inhibition of its target signaling pathway, is able to prolong mouse lifespan, increasing TL [39]. Our findings did not change when excluding these participants.

Although the underlying mechanisms of PCBs on TL are not still clear, three basic mechanisms may explain their toxic effects: [1] they have an ability to bind to specific molecular sites of action (receptors or enzymes, among others), [2] to bind target molecules (DNA and proteins), and [3] to accumulate in lipid-rich tissue and tissue compartments (generally fat and liver) [41]. Furthermore, the mechanism associated with PCBs and dioxins toxicity may rely on their action as ligands of AhR promoting expression of several inflammatory markers and increasing oxidative stress [42,43]. It seems that PCBs could interfere with the cell cycle progression, reducing telomerase activity and thus shortening TL [22].

### Strengths and Limitations

The present study included several strengths. The technique used to measure TL (MMqPCR) allows the quantification of telomere and the single copy gene in the same well in a single reaction, which reduces potential measurement errors. The relative large sample size of the study and the repeated validation of the FFQ in several research works [28] also strengthen the results. Moreover, DNA extracted from saliva samples is a non-invasive and less expensive method that allows larger amounts of DNA from a simple extraction. In addition, we used a variety of sensitivity analyses to support the robustness of the results and we adjusted for a wide range of potential confounders. To strengthen the results shown in this investigation, we used Simes’ method to penalize for multiple comparisons.

However, there are several limitations that should be noted. First, although several previous studies evaluated individual different congeners, we clustered them in groups to assess the total dietary exposure to PCBs, dioxins, and DL-PCBs alone or in combination with dioxins, which may have led to some degree of underestimation in our results. Second, we only assessed POPs exposure from dietary sources, although there are others, and this source of information is acknowledged to be affected by measurement error, which will attenuate the associations, given that it is more likely to induce a non-differential misclassification. However, it is known that, in the general population, the relevant exposure occurs mainly through diet [5] and the assessment of dietary exposures represents therefore a sound approach for the assessment of the effects of these compounds. Third, we have no data on aspects related to food preparation or the cooking methods that may alter POPs levels in the final food product, which may result in an underestimation of POPs intake. Fourth, we did not measure POPs from adipose tissue or blood, and thus the pharmacokinetics and bioavailability of different POPs congeners were not taken into account, which could result in a less accurate estimation of POPs exposure, adding another potential source of non-differential measurement error and misclassification. However, blood concentrations may not be the best measure to represent the bioactive POPs since it is widely known that POPs largely accumulate in adipose tissue during a lifetime and become a source of chronic internal exposure because they are continuously released from adipose tissue to the circulation and to vital organs with lipid content [9]. Fifth, information of POPs exposure was collected from a previous study in a Spanish population [27], and thus we cannot discard temporal-spatial changes in food concentrations of PCBs or dioxins. Sixth, the healthier profile of individuals of the SUN cohort may not represent the general population. However, this limitation may reduce potential confounders related to socioeconomic status and educational level and also contributes to increasing the validity of their self-reported answers. Seventh, even though the DNA isolated from saliva samples contains leukocytes and epithelial cells at varying proportions, and TL is different in cells or tissues, we can assume that salivary TL and leukocyte TL are positively correlated [44]. Lastly, eighth, the design of the study is cross-sectional, and thus, causal effects cannot be inferred.

## 5. Conclusions

This cross-sectional study of individuals over 55 years showed that dietary exposure to total PCBs and DL-PCBs alone and together with dioxins were associated with TL. However, dietary dioxins exposure was not found to have association with TL. Further research studies, preferably with POPs measured in biological samples, are needed to confirm these observations using larger longitudinal studies with repeated measures of TL.

## Figures and Tables

**Figure 1 nutrients-14-00353-f001:**
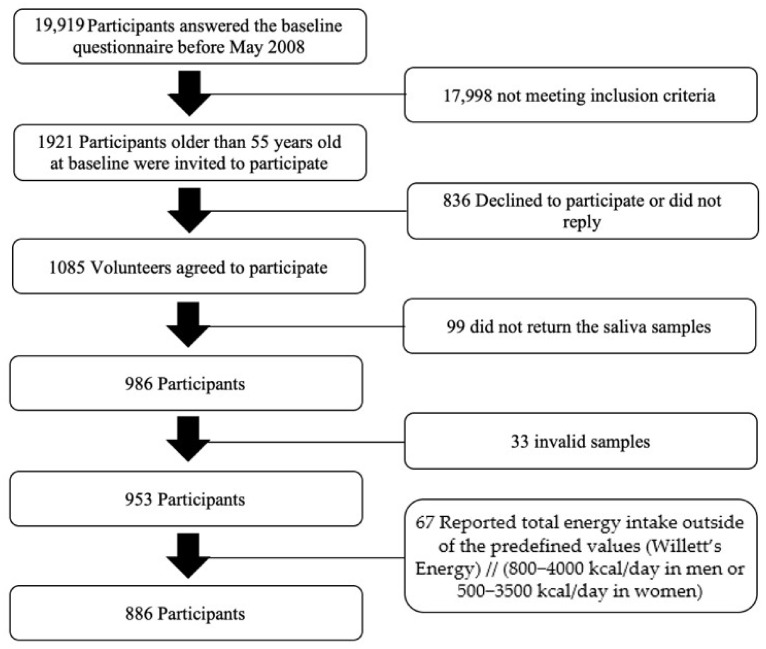
Flowchart of study participants.

**Figure 2 nutrients-14-00353-f002:**
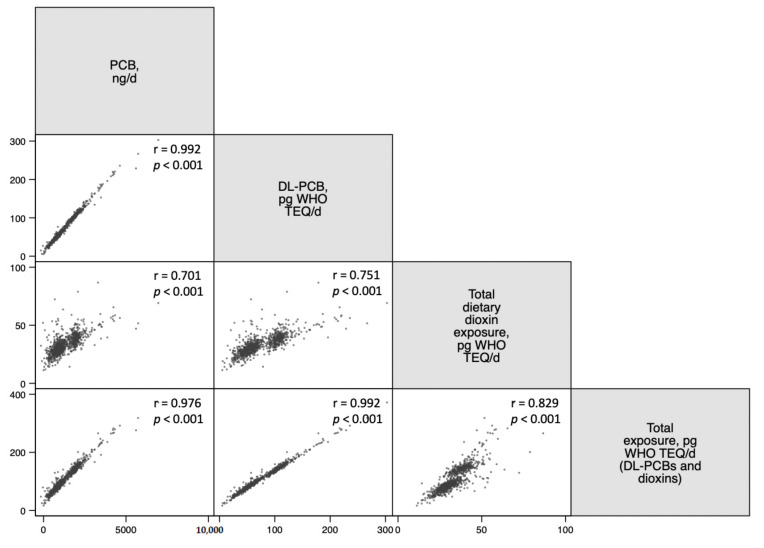
Spearman correlation analysis between dietary exposure to PCBs and dioxins energy-adjusted intake. Abbreviations: DL-PCBs; Dioxin-like polychlorinated biphenyls; PCBs, polychlorinated biphenyls; SUN, Seguimiento Universidad de Navarra; TEQ, Toxic Equivalents; WHO, World Health Organization.

**Table 1 nutrients-14-00353-t001:** Descriptive characteristics of the study participants from the SUN project according to the median of dietary exposure to PCBs and dioxins energy-adjusted intake of participants over 55 years from the SUN project (*n* = 886).

	Overall	PCBs-Energy Adjusted Intake (ng/d)		Dioxins-Energy Adjusted Intake (pg WHO TEQ/d)		Total Exposure to pg WHO TEQ/d (DL-PCBs and Dioxins)	
		<1125.13	≥1125.13	<32.80	≥32.80	<97.80	≥97.80
** *N* **	886	443	443	443	443	433	443
**Males**	645 (73)	328 (74)	317 (72)	330 (74)	315 (71)	350 (79)	295 (67)
**Telomere length, T/S ratio**	0.70/0.69 (0.21)	0.72/0.69 (0.21)	0.69/0.68 (0.19)	0.71/0.69 (0.21)	0.69/0.68 (0.20)	0.71/0.69 (0.20)	0.69/0.69 (0.20)
**Age, y**	67.70 (6.10)	66.91 (6.05)	66.93 (6.16)	66.76 (6.10)	67.08 (6.11)	66.65 (5.94)	67.19 (6.26)
**Body mass index, kg/m^2^**	25.85 (3.15)	25.64 (2.98)	26.06 (3.30)	25.78 (3.08)	25.92 (3.23)	25.66 (3.04)	26.03 (3.26)
**Physical activity, MET-h/week**	22.64 (19.81)	22.06 (20.03)	23.22 (19.59)	23.28 (20.46)	22.01 (19.14)	22.13 (19.75)	23.15 (19.88)
**Total energy intake, kcal/d**	2247.49 (649.44)	2252.88 (683.08)	2242.11 (614.69)	2245.22 (680.33)	2249.76 (617.77)	2253.98 (675.08)	2241.01 (623.44)
**Computer use, h/d**	1.44 (1.55)	1.40 (1.49)	1.48 (1.60)	1.41 (1.44)	1.46 (1.64)	1.42 (1.48)	1.45 (1.61)
**TV watching (<3 h/d)**	87 (9.82)	40 (9.03)	47 (10.61)	43 (9.71)	44 (9.93)	42 (9.48)	45 (10.16)
**Sedentarism, h/d**	4.07 (2.38)	3.92 (2.06)	4.23 (2.65)	4.01 (2.12)	4.14 (2.60)	3.96 (2.06)	4.19 (2.65)
**Night sleeping, h/d**	7.18 (0.75)	7.20 (0.72)	7.16 (0.78)	7.17 (0.72)	7.18 (0.79)	7.21 (0.71)	7.14 (0.79)
**Sleeping siesta, h/d**	0.39 (0.76)	0.39 (0.70)	0.39 (0.81)	0.37 (0.67)	0.41 (0.83)	0.37 (0.64)	0.41 (0.86)
**Years at university, y**	5.34 (1.87)	5.47 (1.92)	5.22 (1.81) *	5.45 (1.87)	5.23 (1.86) *	5.49 (1.90)	5.20 (1.82) *
**Special diet**	134 (15.12)	65 (14.67)	69 (15.58)	62 (14.00)	72 (16.25)	63 (14.22)	71 (16.03)
**Snacking between hours**	185 (20.88)	99 (22.35)	86 (19.41)	102 (23.02)	83 (18.74)	101 (22.80)	84 (18.96)
**Personal history of CVD**	74 (8.35)	36 (8.13)	38 (8.58)	36 (8.13)	38 (8.58)	37 (8.35)	37 (8.35)
**Personal history of hypertension**	463 (52.26)	217 (48.98)	246 (55.53)	219 (49.44)	244 (55.08)	220 (49.66)	243 (54.85)
**Personal history of diabetes**	80 (9.03)	42 (9.48)	38 (8.58)	40 (9.03)	40 (9.03)	40 (9.03)	40 (9.03)
**Mediterranean diet score (0–9)**	4.97 (1.73)	4.53 (1.63)	5.41 (1.71) **	4.65 (1.70)	5.28 (1.70) **	4.45 (1.66)	5.49 (1.63) **
**Smoking status**							
**Current**	133 (15.01)	72 (16.25)	61 (13.77)	71 (16.03)	62 (14.00)	72 (16.25)	61 (13.77)
**Never**	292 (32.96)	144 (32.51)	148 (33.41)	130 (29.35)	162 (36.57) *	136 (30.70)	156 (35.21)
**Former**	459 (51.81)	226 (51.02)	233 (52.60)	241 (54.40)	218 (49.21)	234 (52.82)	225 (50.79)
**Total cholesterol intake, mg/d**	388.03 (121.55)	367.22 (115.38)	408.84 (124.11) **	367.33 (112.55)	408.73 (126.71) **	366.69 (116.54)	409.37 (122.83) **
**Ultra-processed food consumption, servings/d**	2.53 (1.30)	2.72 (1.26)	2.34 (1.32) **	2.83 (1.40)	2.23 (1.12) **	2.81 (1.38)	2.25 (1.15) **
**Total dietary fiber intake, g/d**	30.91 (12.21)	29.57 (12.32)	32.25 (11.97) **	28.19 (10.13)	33.63 (13.46) **	29.29 (11.89)	32.53 (12.32) **
**Alcohol consumption, g/d**	9.89 (13.38)	9.92 (13.66)	9.86 (13.10)	11.41 (15.62)	8.38 (10.48) *	10.47 (14.21)	9.31 (12.49)
**Macro-nutrients intake**							
**Carbohydrate intake, % energy**	43.92 (8.26)	45.29 (8.27)	42.54 (8.02) **	45.65 (7.89)	42.19 (8.27) **	45.52 (8.20)	42.32 (8.01) **
**Protein intake, % energy**	18.67 (3.52)	17.41 (3.34)	19.92 (3.24) **	17.12 (3.13)	20.22 (3.20) **	17.24 (3.22)	20.09 (3.22) **
**Total fat intake, % energy**	34.30 (6.93)	34.15 (7.21)	34.45 (6.64)	33.55 (6.87)	35.04 (6.92) **	33.94 (7.16)	34.65 (6.68)

Values are means (SD) for continuous variables and *n* (%) for categorical variables. CVD, Cardiovascular disease; DL-PCBs; Dioxin-like polychlorinated biphenyls; MET, metabolic equivalent; PCBs, polychlorinated biphenyls; T/S ratio, telomeres and a single copy gene (albumin) ratio; SD, standard deviation; SUN, Seguimiento Universidad de Navarra; TEQ, Toxic Equivalents; WHO, World Health Organization; y, years; d, day; * indicates *p* values < 0.05; ** indicates *p* values < 0.01.

**Table 2 nutrients-14-00353-t002:** Multivariable regression models of relative telomere length (SD of T/S ratio) and dietary exposure to PCBs and dioxins of participants over 55 years from the SUN project.

	Model 1			Model 2			Model 3			Model 4		
	SD of T/S Ratio (95% CI)	*p*-Value	FDR *	SD of T/S Ratio (95% CI)	*p*-Value	FDR *	SD of T/S Ratio (95% CI)	*p*-Value	FDR *	SD of T/S Ratio (95% CI)	*p*-Value	FDR *
**Total PCBs, ng/d**	−0.20 (−0.42 to 0.01)	0.063	0.115	−0.19 (−0.41 to 0.02)	0.081	0.145	−0.21 (−0.43 to 0.005)	0.055	0.096	−0.30 (−0.55 to −0.06)	**0.015**	**0.0347**
**Dioxin-like PCBs, pg WHO TEQ/d**	−4.16 (−8.68 to 0.37)	0.072	0.115	−3.90 (−8.45 to 0.65)	0.093	0.145	−4.36 (−8.95 to 0.23)	0.062	0.096	−6.17 (−11.30 to −1.03)	**0.019**	**0.0347**
**Total Dioxins, pg WHO TEQ/d**	−10.10 (−30.00 to 9.67)	0.315	0.315	−9.51 (−29.50 to 10.50)	0.350	0.350	−11.90 (−32.20 to 8.30)	0.247	0.247	−13.90 (−37.70 to 9.79)	0.249	0.249
**Total exposure to TEQ, pg WHO TEQ/d (DL-PCBs and dioxins)**	−3.36 (−7.19 to 0.47)	0.086	0.115	−3.15 (−7.00 to 0.70)	0.109	0.145	−3.57 (−7.46 to 0.32)	0.072	0.096	−5.02 (−9.44 to −0.61)	**0.026**	**0.0347**

All SD and 95% confidence interval values are ×10^−7^. Model 1: Adjusted for age and sex. Model 2: Further adjusted for BMI (kg/m^2^), energy intake (kcal/day), and personal history of CVD, obesity, HTA, diabetes, cancer, and dyslipidemia (yes or no). Model 3: Further adjusted for educational level (year at university, continuous), smoking status (current, never, former), physical activity (MET-h/week, continuous), computer hours (continuous), TV hours (continuous), sleeping hours (continuous), sleeping/siesta (yes or no), and snacking between hours (yes or no). Model 4: Further adjusted for alcohol consumption (g/day, continuous), cholesterol intake (mg/day), fiber intake (g/day, continuous), total fats intake (percentage of total energy intake of lipids, continuous), ultra-processed food consumption (servings/day, continuous), following a special diet (yes or no), and Mediterranean diet (scale 0–9, continuous). Abbreviations: DL-PCBs; Dioxin-like polychlorinated biphenyls; FDR, False Discovery Rate; MET, metabolic equivalent; T/S ratio, telomeres and a single copy gene (albumin) ratioPCBs, polychlorinated biphenyls; SD, standard deviation; SUN, Seguimiento Universidad de Navarra; TEQ, Toxic Equivalents; WHO, World Health Organization. * Adjustment for multiple comparisons by the Simes method. d, day. Significant values are in bold type.

**Table 3 nutrients-14-00353-t003:** Sensitivity analyses for the association between telomere length (SD of T/S ratio) and dietary exposure to PCBs and dioxins energy-adjusted intake of subjects over 55 years from the SUN project.

	*n*	Total PCB, ng/d		DL-PCBs, pg WHO TEQ/d		Total Dioxins, pg WHO TEQ/d		Total Exposure, pg WHO TEQ/d (DL-PCBs and Dioxins)	
		SD of T/S Ratio (95% CI)	*p*-Value	SD of T/S Ratio (95% CI)	*p*-Value	SD of T/S Ratio (95% CI)	*p*-Value	SD of T/S Ratio (95% CI)	*p*-Value
**Overall (*n* = 886)**	886	−0.30 (−0.55 to −0.06)	**0.015**	−6.17 (−11.30 to −1.03)	**0.019**	−13.90 (−37.70 to 9.79)	0.249	−5.02 (−9.44 to −0.61)	**0.026**
**Energy limits: percentiles 5–95 (*n* = 89)**	797	−0.35 (−0.61 to −0.08)	**0.010**	−7.15 (−12.70 to −1.61)	**0.011**	−20.80 (−46.50 to 5.00)	0.114	−5.98 (−10.70 to −1.22)	**0.014**
**Additionally adjusted for:**									
**Omega-3 fatty acids intake**	886	−0.45 (−0.77 to −0.13)	**0.006**	−9.72 (−16.70 to −2.77)	**0.006**	−14.20 (−41.00 to 12.50)	0.295	−7.67e (−13.60 to −1.78)	**0.011**
**Years elapsed between saliva collection and the inclusion in the study**	886	−0.301 (−0.54 to −0.58)	**0.015**	−6.14 (−11.30 to −1.01)	**0.019**	−13.60 (−37.4 to 10.1)	0.259	−5.00 (−9.40 to −0.59)	**0.026**
**Excluding participants:**									
**With family history of obesity (*n* = 181)**	705	−0.38 (−0.65 to −0.11)	**0.005**	−7.90 (−13.60 to −2.15)	**0.007**	−20.50 (−47.40 to 6.44)	0.136	−6.53 (−11.50 to −1.59)	**0.010**
**With family history of CVD (*n* = 216)**	670	−0.24 (−0.52 to 0.04)	0.093	−4.78 (−10.80 to 1.21)	0.118	−14.20 (−42.40 to 14.00)	0.323	−4.01 (−9.16 to 1.14)	0.127
**With family history of HTA (*n* = 230)**	656	−0.35 (−0.64 to −0.06)	**0.019**	−7.34 (−13.50 to −1.21)	**0.019**	−15.50 (−43.70 to 12.70)	0.281	−5.95 (−11.20 to −0.68)	**0.027**
**With family history of diabetes (*n* = 169)**	717	−0.32 (−0.57 to −0.07)	**0.013**	−6.61 (−11.90 to −1.28)	**0.015**	−10.70 (−36.10 to 14.70)	0.410	−5.23 (−9.81 to −0.65)	**0.025**
**With family history of cancer (*n* = 105)**	781	−0.25 (−0.51 to 0.02)	0.071	−4.77 (−10.40 to 0.85)	0.096	−2.19 (−28.40 to 24.00)	0.870	−3.60 (−8.44 to 1.24)	0.144
**With personal history of obesity (*n* = 128)**	758	−0.28 (−0.54 to −0.02)	**0.033**	−5.62 (−11.10 to −0.13)	**0.045**	−12.10 (−38.10 to 13.80)	0.359	−4.59 (−9.33 to 0.15)	0.058
**With personal history of CVD (*n* = 74)**	812	−0.308 (−0.56 to −0.05)	**0.021**	−6.04 (−11.50 to −0.574)	**0.030**	−9.59 (−34.30 to 15.10)	0.447	−4.78 (−9.47 to −0.09)	**0.046**
**With personal history of HTA (*n* = 463)**	423	−0.46 (−0.86 to −0.05)	**0.027**	−9.76 (−18.10 to −1.40)	**0.022**	−18.10 (−54.30 to 18.10)	0.326	−7.72 (−14.80 to −0.63)	**0.033**
**With personal history of diabetes (*n* = 80)**	806	−0.32 (−0.56 to −0.07)	**0.011**	−6.56 (−11.70 to −1.42)	**0.012**	−15.00 (−38.60 to 8.60)	0.213	−5.35 (−9.75 to −0.94)	**0.017**
**With personal history of dyslipidemia (*n* = 176)**	710	−0.32 (−0.60 to −0.05)	**0.022**	−6.98 (−12.80 to −1.13)	**0.019**	−13.80 (−40.20 to 12.70)	0.307	−5.61 (−10.60 to −0.60)	**0.028**
**With personal history of cancer (*n* = 79)**	807	−0.32 (−0.57 to −0.06)	**0.016**	−6.46 (−11.90 to −1.02)	**0.020**	−13.50 (−38.50 to 11.60)	0.291	−5.20 (−9.86 to −0.55)	**0.029**
**Taking drugs to treat Diabetes and immunosuppressants for alzheimers**	868	−0.249 (−0.48 to −0.02)	**0.037**	−5.10 (−10.10 to −0.15)	**0.043**	−9.87 (−32.70 to 12.90)	0.396	−4.10 (−8.35 to 0.151)	0.059

All SD and 95% confidence interval values are ×10^−7^. Multivariable linear regression models adjusted for age and sex, body mass index (kg/m^2^), energy intake (kcal/day), personal history of CVD, obesity, HTA, diabetes, cancer and dyslipidemia (yes or no), educational level (year at university, continuous), smoking status (current, never, former), physical activity (MET-h/week, continuous), computer hours (continuous), TV hours (continuous), sleeping hours (continuous), sleeping/siesta (yes or no), snacking between hours (yes or no), alcohol consumption (g/day, continuous), cholesterol intake (mg/day), fiber intake (g/day, continuous), total fats intake (percentage of total energy intake of lipids, continuous), ultra-processed food consumption (servings/day, continuous), following a special diet (yes or no), and Mediterranean diet (scale 0–9, continuous). DL-PCBs; Dioxin-like polychlorinated biphenyls; MET, metabolic equivalent; T/S ratio, telomeres and a single copy gene (albumin) ratio; PCBs, polychlorinated biphenyls; standard deviation, SD; SUN, Seguimiento Universidad de Navarra; TEQ, Toxic Equivalents; WHO, World Health Organization. d, day. Significant values are in bold type.

## Data Availability

SUN Project: ClinicalTrials.gov Identifier: NCT02669602. Available online: https://clinicaltrials.gov/ct2/show/NCT02669602 (accessed on 12 January 2022).

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
