# Peer review of "Dietary Exposure to Polychlorinated Biphenyls and Dioxins and Its Relationship to Telomere Length in Subjects Older Than 55 Years from the SUN Project"

_nutrients, 2022, doi:10.3390/nu14020353_

Round 1

Reviewer 1 Report

The paper by Alonso-Pedrero et al., presents results obtained by analyzing the same SUN cohort, which showed shorter telomere lengths to a Mediterranean diet as well as to cardiovascular health and ultra-processed foods amongst others in the same 886 individuals. I have some major concerns about the this paper as written.

Major Concerns

  1. If polychlorinated biphenyls (PCBs) have previously been shown to cause cancer and thereby activate the telomerase enzyme which elongates telomeres, is there any concerns as to why this study shows shorter telomere lengths to PCBs?  
  2. Could this study be identifying a correlation between the ultra-processed food  and shorter telomere lengths since the paper as written does indicate that 812 of the 886 individuals have a personal history of CVD as well as 667 of the 882 individuals have a personal history of hypertension?
  3. On line 99, a food frequency questionnaire (FFQ) was used to evaluate dietary intake of food but there is no mention of how the PCBs, DL-PCBs, total dioxins and total exposure (DL-PCBs and dioxins) were estimated as the questionnaire only assess dietary nutritional values. This number is then expressed as a TEQ and is then used to determine a TEF and finally multiplied by the concentration consumed. Even though the authors address why blood may not be the most viable option to measure POP exposure, the employed estimation protocol does not seem as founded as analyzing serum PCB levels in the  participants for a more accurate measurement compared to the recall bias questionnaire estimation protocol? 
  4. There is no mention of how the T/S ratio was calculated even in the citation used. Please use the Telomere Research Network minimum reporting guidelines when publishing results to address this and additional questions concerning the telomere length results. https://osf.io/9pzst/
  5. In table 1, the heading lists using the median values for dietary exposure to PCBs and dioxin energy values but the footnote under this table lists using the means (SD) for continuous variables.  This occurs in the text (Line 100) too.  Did the analysis analyze the mean or median values?
  6. Line 150-152 says "We used SD (Standard Deviation) of T/S ratio because showing that there is a change of such magnitude for each SD of the independent variable gives a better idea of the effect." Using SD is dependent on the individuals selected for each group and how disperse the telomere length is from the mean/median.  Since it is difficult to see the mean/median of the groups being analyzed, please consider adding the mean/median telomere length result and SD to Table 1.
  7. Since it is well established that women have longer telomere lengths than men when adults, it would be helpful to see what is the number of females and males included in the various analysis groups mentioned in Table 1. Please consider adding this information to Table 1.
  8. Line 152-164: Please consider explaining what was the criteria for including participants into the four different models. How are these included in Table 1, which lists PCBs-energy adjusted intake and Dioxins-energy adjusted intake? Are these the same as that listed on line 177 which says "median of dietary PCBs, dioxins and total exposure?   Please use consistent labeling throughout the manuscript if these are the same.
  9. Please define what DL-PCBs (dioxin-like PCBs) is when first used on line 100 rather than see it define in Figure 2.  How is this now determined and calculated?
  10. Similarly, what is TCDD and how is this calculated?
  11. In the supplementary table 1, a) what was the criteria used to determine the value used to segregate Total PCB, DL-PCBs and Total dioxin into the two columns for the four models listed. b) Why is there a 1 (ref) only listed for the first column for the Total PCB; DL-PCBs and Total dioxin columns? c) What is the power estimates to show there is sufficient power for the analysis if looking at < 20th  percentile for short telomeres in the four models listed for the two different columns for Total PCB; DL-PCBs and Total dioxin?
  12. How has this study addressed individuals taking the telomere lengthening medical drugs to treat Diabetes (metformin) and immunosuppressants (rapamycin) for Alzheimers and Huntington's disease?

Minor Comments

  1. Line 86-94: Duplicated information in the text and in Figure 1. Consider only using one of these.
  2. If investigating PBCs and dioxin, why does the title only mention PBCs and telomere length from the SUN project?
  3. All sub headings in the Materials and Methods sections begin with a 2.1. Consider renumbering these sub headings.

Author Response

Comments from the Editors and Reviewers:

Manuscript Reference:  nutrients-1499656

Title: Dietary exposure to polychlorinated biphenyls and dioxins, and its relationship with telomere length in subjects older than 55 years old from the SUN project

Section: Nutrition and Public Health

Special Issue: Public Health Nutrition and Healthy Aging

Submission Date: 23 November 2021

We thank the Editorial Office for their thoughtful comments. We have carefully revised the manuscript and addressed all of the suggestions of the reviewers. Please find here below our answers to the referee’s comments.

*All pages and lines referenced correspond to manuscript with changes highlighted in yellow.

Comments and Suggestions for Authors

The paper by Alonso-Pedrero et al., presents results obtained by analyzing the same SUN cohort, which showed shorter telomere lengths to a Mediterranean diet as well as to cardiovascular health and ultra-processed foods amongst others in the same 886 individuals. I have some major concerns about the this paper as written.

Major Concerns

If polychlorinated biphenyls (PCBs) have previously been shown to cause cancer and thereby activate the telomerase enzyme which elongates telomeres, is there any concerns as to why this study shows shorter telomere lengths to PCBs?  

Thank you for your remark. It is certainly true that telomeres and telomerase have been shown to be strongly involved in the carcinogenic process. In several studies it is reported that telomerase is not essential for oncogenic transformation or tumor formation inasmuch as the shortening of telomeres produce a very high likelihood of telomere fusions and genomic instability associated with the development of cancer [1,2]. Therefore, even though this study showed telomere shortening, it does not mean that could not cause cancer. Moreover, several studies have linked exposure to PCBs with a reduction in telomerase activity [3].

  1. Senthilkumar PK, Klingelhutz AJ, Jacobus JA, Lehmler H, Robertson LW, Ludewig G. Airborne polychlorinated biphenyls (PCBs) reduce telomerase activity and shorten telomere length in immortal human skin keratinocytes (HaCat). Toxicol Lett. 2011;204:64-70. doi:10.1016/j.toxlet.2011.04.012.
  2. Blasco MA, Lee HW, Hande MP, et al. Telomere shortening and tumor formation by mouse cells lacking telomerase RNA. Cell. 1997;91:25-34. doi:10.1016/s0092-8674(01)80006-4.
  3. Senthilkumar, P.K.; Klingelhutz, A.J.; Jacobus, J.A.; Lehmler, H.; Robertson, L.W.; Ludewig, G. Airborne polychlorinated biphenyls (PCBs) reduce telomerase activity and shorten telomere length in immortal human skin keratinocytes (HaCat). Toxicol. Lett. 2011, 204, 64–70, doi:10.1016/j.toxlet.2011.04.012.

Could this study be identifying a correlation between the ultra-processed food and shorter telomere lengths since the paper as written does indicate that 812 of the 886 individuals have a personal history of CVD as well as 667 of the 882 individuals have a personal history of hypertension?

Yes, indeed we have recently published an article associating ultra-processed food consumption and the risk of short telomeres in this population of adults older than 55 years old from the SUN project [1], which is highly linked with individuals with personal history of CVD and personal history of hypertension. However, our results are independent of ultra-processed food consumption since we have adjusted for it in the multivariable analysis.

  1. Alonso-Pedrero, L.; Ojeda-Rodríguez, A.; Martínez-González, M.A.; Zalba, G.; Bes-Rastrollo, M.; Marti, A. Ultra-processed food consumption and the risk of short telomeres in an elderly population of the Seguimiento Universidad de Navarra ( SUN ) Project. Am. J. Clin. Nutr. 2020, 1–8, doi:10.1093/ajcn/nqaa075.

On line 99, a food frequency questionnaire (FFQ) was used to evaluate dietary intake of food but there is no mention of how the PCBs, DL-PCBs, total dioxins and total exposure (DL-PCBs and dioxins) were estimated as the questionnaire only assess dietary nutritional values. This number is then expressed as a TEQ and is then used to determine a TEF and finally multiplied by the concentration consumed. Even though the authors address why blood may not be the most viable option to measure POP exposure, the employed estimation protocol does not seem as founded as analyzing serum PCB levels in the participants for a more accurate measurement compared to the recall bias questionnaire estimation protocol? 

Thank you for your appreciation, we clarified this information in lines 99-104. According to the second part of the sentence, we totally agree with the reviewer. However, the FFQ of the SUN project was not designed to estimate PCBs and dioxins and for sure, as you mentioned, having measurements in blood would allow more precise measurements.

There is no mention of how the T/S ratio was calculated even in the citation used. Please use the Telomere Research Network minimum reporting guidelines when publishing results to address this and additional questions concerning the telomere length results. https://osf.io/9pzst/

Thank you for your comment. We have described it in more detail following the guidelines of the Telomere Research Network website [1]. (Lines 120-130)

  1. Telomere Research Network. Available online: https://trn.tulane.edu/resources/study-design-analysis/ (accessed on 10 December 2021).

In table 1, the heading lists using the median values for dietary exposure to PCBs and dioxin energy values but the footnote under this table lists using the means (SD) for continuous variables.  This occurs in the text (Line 100) too.  Did the analysis analyze the mean or median values?

In this table (table 1) we analyzed the descriptive characteristics of the study participants above and below the median values for dietary exposure to PCBs, dioxins and total exposure. The median values for dietary exposure to PCBs and dioxin are used as a threshold to evaluate participants above and below the dietary exposure to the POPs evaluated. Meanwhile, mean values with SD in parentheses are the results of the main characteristics evaluated in the population for continuous variables.

Line 150-152 says "We used SD (Standard Deviation) of T/S ratio because showing that there is a change of such magnitude for each SD of the independent variable gives a better idea of the effect." Using SD is dependent on the individuals selected for each group and how disperse the telomere length is from the mean/median.  Since it is difficult to see the mean/median of the groups being analyzed, please consider adding the mean/median telomere length result and SD to Table 1.

Thank you for your comment. We have added mean/median values of telomere length into Table 1.

Since it is well established that women have longer telomere lengths than men when adults, it would be helpful to see what is the number of females and males included in the various analysis groups mentioned in Table 1. Please consider adding this information to Table 1.

Thank you for your comment. We have added the number and percentage of men in table 1.

Line 152-164: Please consider explaining what was the criteria for including participants into the four different models. How are these included in Table 1, which lists PCBs-energy adjusted intake and Dioxins-energy adjusted intake? Are these the same as that listed on line 177 which says "median of dietary PCBs, dioxins and total exposure?   Please use consistent labeling throughout the manuscript if these are the same.

In table 1 we wanted to include the main variables used to adjust the analyses of this paper. We adjusted for different confounding factors considering the existing bibliography. Model 1 was only adjusted for age and sex; model 2 was focus on personal history of several diseases; model 3 on lifestyle factors and model 4 on diet.

According to the second question, it is not the same PCBs-energy adjusted intake and Dioxins-energy adjusted intake and median of dietary PCBs, dioxins and total exposure. PCBs-energy adjusted intake and Dioxins-energy adjusted intake refers to the entire variable and median of dietary PCBs, dioxins and total exposure divide the population in subjects above and below the median of intake refers to the entire variable and median of dietary

Please define what DL-PCBs (dioxin-like PCBs) is when first used on line 100 rather than see it define in Figure 2.  How is this now determined and calculated?

Thank you for your comment. We have defined it in line 99. We calculated DL-PCBs by multiplying the average concentrations in several food items obtained from a previous study [1] with the respective frequency of consumption (in grams).

  1. Llobet, J.M.; Martí-Cid, R.; Castell, V.; Domingo, J.L. Significant decreasing trend in human dietary exposure to PCDD/PCDFs and PCBs in Catalonia, Spain. Toxicol. Lett. 2008, 178, 117–26, doi:10.1016/j.toxlet.2008.02.012.

Similarly, what is TCDD and how is this calculated?

Thank you for your comment. As we did not calculate TCDD since that information was obtained from a previous study [1], we deleted that from the manuscript.

  1. Llobet, J.M.; Martí-Cid, R.; Castell, V.; Domingo, J.L. Significant decreasing trend in human dietary exposure to PCDD/PCDFs and PCBs in Catalonia, Spain. Toxicol. Lett. 2008, 178, 117–26, doi:10.1016/j.toxlet.2008.02.012.

In the supplementary table 1, a) what was the criteria used to determine the value used to segregate Total PCB, DL-PCBs and Total dioxin into the two columns for the four models listed. b) Why is there a 1 (ref) only listed for the first column for the Total PCB; DL-PCBs and Total dioxin columns? c) What is the power estimates to show there is sufficient power for the analysis if looking at < 20th percentile for short telomeres in the four models listed for the two different columns for Total PCB; DL-PCBs and Total dioxin?

a. The criteria used to segregate total PCB, DL-PCBs and Total dioxin into the two columns is above and below the median value as we did in table 1.

b. We did logistic regression models to determine the risk of short telomeres below the 20th percentile considering the first category (below the median) as the reference category giving it the value 1.

c. Thank you for your comment. We agree with you that our sample size was not large enough to detect significant differences in the risk of telomere shortening between the groups (above/below the median of PCBs, DL-PCBs and dioxins). In addition, as you pointed out, we have not looked into the penalty for multiple comparisons, which may also minimize the p-values. Therefore, we have decided to omit this table into the revised version of the manuscript (supplementary table 1). 

How has this study addressed individuals taking the telomere lengthening medical drugs to treat Diabetes (metformin) and immunosuppressants (rapamycin) for Alzheimers and Huntington's disease?

We do know that Rapamycin, through the inhibition of the TOR signaling pathway, is able to prolong mouse lifespan, just as lifespan of diverse animal species; and Metformin does attenuate the hallmarks of aging including telomere erosion. Unfortunately, due to the low number of subjects taken these drugs in this study (18 subjects), we did not take this issue into consideration. However, we have included this information in the sensitivity analyses excluding participants taking these drugs (Table 3).

Minor Comments

Line 86-94: Duplicated information in the text and in Figure 1. Consider only using one of these.

We prefer to explain this information in the legend of figure 1 and in the text. We consider that it is necessary.

If investigating PBCs and dioxin, why does the title only mention PBCs and telomere length from the SUN project?

Thank you for your comment. As the paper only show significant results with PCBs, we wanted to emphasize it in the title. However, we are including them in the title.

Now it reads as follows: Dietary exposure to polychlorinated biphenyls and dioxins, and its relationship with telomere length in subjects older than 55 years old from the SUN project

All sub headings in the Materials and Methods sections begin with a 2.1. Consider renumbering these sub headings.

Thank you for your comment. We have renumbered the sub headings properly.

Reviewer 2 Report

The manuscript nutrients-1499656 entitled “Dietary exposure to polychlorinated biphenyls and telomere length in subjects over 55 years old from the SUN project” deals with the association of telomere length from salivary samples and PCBs that were assessed using validated FFQs.

The manuscript is within the scope of the journal and generally well written, however the manuscript has several issues that need to be addressed.

Although the part describing strengths and limitations is well written, I suggest expanding the discussion a bit. Based on results from Table 1 and 3, could you make some comments or recommendations how to modulate PCB exposure by choosing or avoiding certain food items? The results after excluding some volunteers from the analysis would also be worthy to discuss.

Add “total PCB exposure” to concluding sentence of abstract, since this was also stated in conclusion part of the manuscript.

Is the TL presented as SD of T/S score or TL? See text in Table 2 and lines 30-33, and 203-205.

Line 227 Where is a box plot presented?

There are leftovers from guide for authors (Line 234-236).

Change some minor errors: superscript -7 (lines 30-33), italic in vitro (lines 255, 267), spaces (lines 301, 314, 330), and please check the rest of the manuscript.

Author Response

Comments from the Editors and Reviewers:

Manuscript Reference:  nutrients-1499656

Title: Dietary exposure to polychlorinated biphenyls and dioxins, and its relationship with telomere length in subjects older than 55 years old from the SUN project

Section: Nutrition and Public Health

Special Issue: Public Health Nutrition and Healthy Aging

Submission Date: 23 November 2021

We thank the Editorial Office for their thoughtful comments. We have carefully revised the manuscript and addressed all of the suggestions of the reviewers. Please find here below our answers to the referee’s comments.

*All pages and lines referenced correspond to manuscript with changes highlighted in yellow.

Comments and Suggestions for Authors

The manuscript nutrients-1499656 entitled “Dietary exposure to polychlorinated biphenyls and telomere length in subjects over 55 years old from the SUN project” deals with the association of telomere length from salivary samples and PCBs that were assessed using validated FFQs.

The manuscript is within the scope of the journal and generally well written, however the manuscript has several issues that need to be addressed.

Although the part describing strengths and limitations is well written, I suggest expanding the discussion a bit. Based on results from Table 1 and 3, could you make some comments or recommendations how to modulate PCB exposure by choosing or avoiding certain food items? The results after excluding some volunteers from the analysis would also be worthy to discuss.

Thank you for your recommendation, we have added it to the main discussion.

Now text reads as follows:

“Diet is the main route of PCBs, dioxins and other POPs for human exposure (90%) [38]. We observed that fish was the main food group contributing to PCBs and dioxin exposure. Recent epidemiological studies found that omega-3 fatty acids consumption (which is the main source of the fish) is able to reduce cardiovascular mortality [3]. However, PCBs exposure (which main food contributor is fish) is linked to cardiovascular mortality [11]. This is a controversial scenario and the justification could be that the benefits of fish itself seem to overcome the negative effects of PCBs. Moreover, the beneficial effect is observed for moderate consumption of fish, included in a Mediterranean diet-based pattern, which, in addition, is accompanied by other food items that can also counteract the effects of POPs on TL, such as fruits and vegetables. Notably, we did not observed changes in our results when we adjusted for consumption of omega-3 fatty acids.

We also excluded participants with family and personal history of several diseases as they also affect the homeostasis of TL in humans. In this regard, we excluded participants taking drugs to treat diabetes (metformin) and immunosuppressants (rapamycin).  As it is known Metformin does attenuate the hallmarks of aging including telomere erosion [40]; meanwhile Rapamycin, which provide neuroprotection for neurodegenerative diseases (Alzheimer’s, Parkinson’s, Huntington’s and spinocerebellar ataxia type 3), through the inhibition of its target signaling pathway, is able to prolong mouse lifespan increasing TL [39]. Our findings did not change when excluding these participants.”

Add “total PCB exposure” to concluding sentence of abstract, since this was also stated in conclusion part of the manuscript.

Thank you for your comment. We have added this information in line 34-35 of the abstract.

Is the TL presented as SD of T/S score or TL? See text in Table 2 and lines 30-33, and 203-205.

Thank you for your comment. TL in this study is expressed as SD of relative ratio of telomere to a single-copy gene (human albumin) (T/S ratio).

Line 227 Where is a box plot presented?

Thank you for your appreciation. The box plot is presented in (Supplementary Figure S3). However, we have deleted that sentence in line 227 because is duplicated in line 204-207.

There are leftovers from guide for authors (Line 234-236).

Change some minor errors: superscript -7 (lines 30-33), italic in vitro (lines 255, 267), spaces (lines 301, 314, 330), and please check the rest of the manuscript.

Thank you. Done. We have carefully also checked the rest of the manuscript.

Round 2

Reviewer 1 Report

The reviewers have adequately addressed some of the previous concerns but there are still some left unanswered.  

  1. Thank you for including the mean/median values into Table 1. Looking at these values, I still am not convinced analyzing the telomere length standard deviation is adequate since there is little differences between the groups with the authors agreeing they are analyzing small samples sizes. Please address how the results seen are not spurious significant results with PCBs. Power estimates would enhance answering this question.
  2. The review of the previous article still has not adequately addressed some of the minimum guidelines as outlined from the Telomere Research Network (10.31219/osf.io/9pzst) for telomere length publication purposes:

a) How was the DNA extracted from the Oragene kit?

b) How was this DNA stored before the telomere length assay after being extracted?

c) What was the method of documenting the DNA quality and integrity was suitable for the telomere length assay and how many samples were analyzed with this method?

d) Source of the mastermix used for the telomere length assay: manufacturer or homemade?

e) What single copy gene was used for the MMPCR method; primer sequences and concentration?

f) PCR efficiency of single copy gene and telomere primers?

g) What was the source of the reference DNA and concentration of control samples and standard curve?

h) What was the analytic method, considering replicate measurements, to determine final telomere length?

i) Method for accounting for variation between sample replicates?

j) Method for accounting for well position effects within plates?

k) Method for accounting for between plate effects?

l) What is the inter-plate and intra-plate variation seen with this reference DNA?

m) % of samples repeated and % sample failing final QC and excluded from further analysis

n) Acceptable range of PCR efficiency for the single copy gene and telomere primers?

Minor points

Line 122: (the Cawthorn method) is not an acceptable reference. Please add the full reference for this paper.  Do you also mean Cawthon not Cawthorn paper?

Author Response

Comments from the Editors and Reviewers:

Manuscript Reference:  nutrients-1499656

Title: Dietary exposure to polychlorinated biphenyls and dioxins, and its relationship with telomere length in subjects older than 55 years old from the SUN project

Section: Nutrition and Public Health

Special Issue: Public Health Nutrition and Healthy Aging

Submission Date: 23 November 2021

We thank the Editorial Office for their thoughtful comments. We have carefully revised the manuscript and addressed all of the suggestions of the reviewers. Please find here below our answers to the referee’s comments.

*All pages and lines referenced correspond to manuscript with changes highlighted in yellow.

Comments and Suggestions for Authors

The reviewers have adequately addressed some of the previous concerns but there are still some left unanswered.  

  1. Thank you for including the mean/median values into Table 1. Looking at these values, I still am not convinced analyzing the telomere length standard deviation is adequate since there is little differences between the groups with the authors agreeing they are analyzing small samples sizes. Please address how the results seen are not spurious significant results with PCBs. Power estimates would enhance answering this question.

In our research work (tables 2 to 3) the main variable is telomere length and number of participants is the complete sample including 886 subjects. When we distributed them according to the median, there are two groups of 443 participants but that is only used in table 1 to show differences in the characteristics of the population. We had small sample sizes when we analyzed the risk of short telomeres in our population, but that information was in Supplementary Table 1, and it was already deleted.

We used standard deviation as a way of interpreting beta values since standard deviation is independent of the units used for the measurement of the independent variable, and therefore saying that there is a change of such magnitude for each standard deviation of the Independent variable gives a better idea of the effect. Anyway, here we attach a new table with the beta values of the analyses.

The power estimates of our analysis is 99.3%. Moreover, once we penalized by the SIMES method, the results are still significant and therefore the results seen are not spurious significant (table 2).

Multivariable regression models of relative telomere length (T/S ratio) and dietary exposure to PCBs and dioxins of participants over 55 y from the SUN project.

Model 1

Model 2

Model 3

Model 4

b (95%CI)

p-Value

% of predicted change

b (95%CI)

p-Value

% of predicted change

b (95%CI)

p-Value

% of predicted change

b (95%CI)

p-Value

% of predicted change

Total PCBs, ng/d

-1.58 (-3.25 to 0.08)

0.063

0.002

-1.49 (-3.17 to 0.18)

0.081

0.002

-1.65 (-3.35 to 0.037)

0.055

0.002

-2.35 (-4.23 to -0.46)

0.015

0.003

     Dioxin-like PCBs, pg WHO TEQ/d

-32.28 (-67.39 to 2.84)

0.072

0.039

-30.25 (-65.55 to 5.04)

0.093

0.037

-33.84 (-69.43 to 1.76)

0.062

0.041

-47.88 (-87.79 to -7.96)

0.019

0.058

Total Dioxins, pg WHO TEQ/d

-78.76 (-232.55 to 75.02)

0.315

0.095

-73.82 (-228.74 to 81.11)

0.350

0.089

-92.63 (-249.7 to 64.45)

0.247

0.112

-108.20 (-294.40 to 76.01)

0.249

0.131

Total exposure to TEQ, pg WHO TEQ/d (DL-PCBs and dioxins)

-26.06 (-55.77 to 3.66)

0.086

0.032

-24.43 (-54.31 to 5.46)

0.109

0.030

-27.72 (-57.88 to 2.45)

0.072

0.034

-38.99 (-73.24 to -4.74)

0.026

0.047

All b coefficients and 95% Confidence Interval values are ´10-5.

Model 1: Adjusted for age and sex.

Model 2: Further adjusted for BMI (kg/m2), energy intake (kcal/d) and personal history of CVD, obesity, HTA, diabetes, cancer and dyslipidemia (yes or no).

 Model 3: Further adjust for educational level (year at university, continuous), smoking status (current, never, former), physical activity (MET-h/week, continuous), computer hours (continuous), TV hours (continuous), sleeping hours (continuous), sleeping/siesta (yes or no), snacking between hours (yes or no).

Model 4: Further adjusted for alcohol consumption (g/d, continuous), cholesterol intake (mg/d), fiber intake (g/d, continuous), total fats intake (percentage of total energy intake of lipids, continuous), ultra-processed food consumption (servings/day, continuous), following a special diet (yes or no) and Mediterranean diet (scale 0-9, continuous).

Abbreviations: DL-PCBs; Dioxin-like Polychlorinated biphenyls; MET, metabolic equivalent; PCBs, Polychlorinated biphenyls; SUN, Seguimiento Universidad de Navarra; TEQ, Toxic Equivalents; WHO, World Health Organization.

Table 2. Multivariable regression models of relative telomere length (SD of ratio T/S) and dietary exposure to PCBs and dioxins of participants over 55 y from the SUN project.

Model 1

Model 2

Model 3

Model 4

SD of ratio T/S (95%CI)

p-Value

FDR*

SD of ratio T/S (95%CI)

p-Value

FDR*

SD of ratio T/S (95%CI)

p-Value

FDR*

SD of ratio T/S (95%CI)

p-Value

FDR*

Total PCBs, ng/d

-0.20  (-0.42 to 0.01)

0.063

0.115

-0.19 (-0.41 to 0.02)

0.081

0.145

-0.21 (-0.43 to 0.005)

0.055

0.096

-0.30 (-0.55 to -0.06)

0.015

0.0347

     Dioxin-like PCBs, pg WHO TEQ/d

-4.16 (-8.68 to 0.37)

0.072

0.115

-3.90 (-8.45 to 0.65)

0.093

0.145

-4.36 (-8.95 to 0.23)

0.062

0.096

-6.17 (-11.30 to -1.03)

0.019

0.0347

Total Dioxins, pg WHO TEQ/d

-10.10 (-30.00 to 9.67)

0.315

0.315

-9.51 (-29.50 to 10.50)

0.350

0.350

-11.90 (-32.20 to 8.30)

0.247

0.247

-13.90 (-37.70 to 9.79)

0.249

0.249

Total exposure to TEQ, pg WHO TEQ/d (DL-PCBs and dioxins)

-3.36 (-7.19 to 0.47)

0.086

0.115

-3.15 (-7.00 to 0.70)

0.109

0.145

-3.57 (-7.46 to 0.32)

0.072

0.096

-5.02 (-9.44 to -0.61)

0.026

0.0347

All SD and 95% Confidence Interval values are ´10-7. Model 1: Adjusted for age and sex. Model 2: Further adjusted for BMI (kg/m2), energy intake (kcal/d) and personal history of CVD, obesity, HTA, diabetes, cancer and dyslipidemia (yes or no). Model 3: Further adjusted for educational level (year at university, continuous), smoking status (current, never, former), physical activity (MET-h/week, continuous), computer hours (continuous), TV hours (continuous), sleeping hours (continuous), sleeping/siesta (yes or no), snacking between hours (yes or no). Model 4: Further adjusted for alcohol consumption (g/d, continuous), cholesterol intake (mg/d), fiber intake (g/d, continuous), total fats intake (percentage of total energy intake of lipids, continuous), ultra-processed food consumption (servings/day, continuous), following a special diet (yes or no) and Mediterranean diet (scale 0-9, continuous). Abbreviations: DL-PCBs; Dioxin-like Polychlorinated biphenyls; FDR, False Discovery Rate; MET, metabolic equivalent; PCBs, Polychlorinated biphenyls; SD, Standard Deviation; SUN, Seguimiento Universidad de Navarra; TEQ, Toxic Equivalents; WHO, World Health Organization. *Adjustment for multiple comparisons by the Simes method.

  1. The review of the previous article still has not adequately addressed some of the minimum guidelines as outlined from the Telomere Research Network (10.31219/osf.io/9pzst) for telomere length publication purposes:

  1. a) How was the DNA extracted from the Oragene kit? DNA was extracted from samples using the Oragene DNA extraction protocol. https://www.dnagenotek.com/ROW/support/collection-instructions/oragene-dna/OG-250.html
  2. b) How was this DNA stored before the telomere length assay after being extracted? It was stored at -20 C after being extracted.
  3. c) What was the method of documenting the DNA quality and integrity was suitable for the telomere length assay and how many samples were analyzed with this method? DNA was considered pure when the A260/280 was greater than 1.80 and A260/230 greater than 2.0. All the samples were analyzed with this method.
  4. d) Source of the mastermix used for the telomere length assay: manufacturer or homemade? It was a manufacturer mastermix (QuantiTect SYBR Green PCR kit (Qiagen).
  5. e) What single copy gene was used for the MMPCR method; primer sequences and concentration? Albumin, sequence concentration. The primer pair for the single-copy gene albuand albd(final concentration 900 nM each).

The primer sequences were albu (5′- CGGCGGCGGGCGGCGCGGGCTGGGCGGAAATGCTGCACAGAATCCTTG-3′) and albd (5′-GCCCGGCCCGCCGCGCCCGTCCCGCCGGAAAAGCATGGTCGCCTGTT-3′)

  1. f) PCR efficiency of single copy gene and telomere primers?

PCR efficiency was 108.16% for telomere primer and 90% for single copy gene primer.

  1. g) What was the source of the reference DNA and concentration of control samples and standard curve? DNA was extracted with the use of a DNA blood extraction kit (Pure Link Genomic DNA, Invitrogen). A calibration curve with a reference DNA sample (150–2.34 ng/µL in 2-fold dilutions) was included in each 384-well plate and used for the relative quantification.
  2. h) What was the analytic method, considering replicate measurements, to determine final telomere length? We calculate the media value for each triplicate measure.
  3. i) Method for accounting for variation between sample replicates? We rerun all the samples with lack of concordant values among triplicates.
  4. j) Method for accounting for well position effects within plates? We did not detect any important contribution of well position effects within plates.
  5. k) Method for accounting for between plate effects? We did not observe any differences due to plate effects.
  6. l) What is the inter-plate and intra-plate variation seen with this reference DNA?

The reliability of our assay was assessed by calculating the intraclass coefficient (ICC) of triplicate measures (T/S ratios).

Both the inter-assay (IRCs run over multiple qPCR plates) and intra-assay ICC (based on all measures) was calculated using the available online R script on the Telomere Research Network website [1]. The inter-assay ICC was 0.956 and the intra-assay ICC was 0.743.

  1. Telomere Research Network. Available online: https://trn.tulane.edu/resources/study-design-analysis/
  2. m) % of samples repeated and % sample failing final QC and excluded from further analysis

1.12% of the samples were repeated and 3.34% were excluded from further analysis.

  1. n) Acceptable range of PCR efficiency for the single copy gene and telomere primers?

The acceptable range of PCR efficiency for the single copy gene and telomere primers was between 90 and 110%.

Reviewer 2 Report

The revised manuscript nutrients-1499656 entitled “Dietary exposure to polychlorinated biphenyls and dioxins, and its relationship with telomere length in subjects older than 55 years old from the SUN project” is now improved, however really minor details remain, so please address them.

Just to be clear the data are presented as SD of T/S ratio (line 125). However, in lines 30-33, 204-206 it is presented as SD of TL/, in Table 1 is ratio T/S (mean median and SD), and in Tables 2 and 3 SD of ratio T/S. Please unify.

Although you have declared checking the rest of the manuscript, there are still some minor details that need to be corrected: missing ° (line 121) at -80°C, superscript “2” (line 130), “did not observe” (line 277).

Author Response

Comments from the Editors and Reviewers:

Manuscript Reference:  nutrients-1499656

Title: Dietary exposure to polychlorinated biphenyls and dioxins, and its relationship with telomere length in subjects older than 55 years old from the SUN project

Section: Nutrition and Public Health

Special Issue: Public Health Nutrition and Healthy Aging

Submission Date: 23 November 2021

We thank the Editorial Office for their thoughtful comments. We have carefully revised the manuscript and addressed all of the suggestions of the reviewers. Please find here below our answers to the referee’s comments.

*All pages and lines referenced correspond to manuscript with changes highlighted in yellow.

Comments and Suggestions for Authors

The revised manuscript nutrients-1499656 entitled “Dietary exposure to polychlorinated biphenyls and dioxins, and its relationship with telomere length in subjects older than 55 years old from the SUN project” is now improved, however really minor details remain, so please address them.

Just to be clear the data are presented as SD of T/S ratio (line 125). However, in lines 30-33, 204-206 it is presented as SD of TL/, in Table 1 is ratio T/S (mean median and SD), and in Tables 2 and 3 SD of ratio T/S. Please unify.

Thank you. We have unified them as SD of T/S ratio.

Although you have declared checking the rest of the manuscript, there are still some minor details that need to be corrected: missing ° (line 121) at -80°C, superscript “2” (line 130), “did not observe” (line 277).

Thank you, done.
